DATA RELEASE

# Arbovirus vectors in municipalities with a high risk of dengue in Cauca, Southwestern Colombia

Catalina Marceló-Díaz[1,*], Carlos Andres Morales[2], Maria Camila Lesmes[1,3], Patricia Fuya[1], Sergio Andres Mendez[1], Horacio Cadena[4], Alvaro Ávila-Díaz[3] and Erika Santamaria[1]

1  Grupo de Entomología, Instituto Nacional de Salud, Avenida Calle 26 51-20, Bogotá, Colombia
2  Secretaría de Salud Departamental del Cauca, Calle 5 15-57 Barrio Valencia, Popayán, Colombia
3  Universidad de Ciencias Aplicadas y Ambientales, Calle 222 55-37, Bogotá, Colombia
4  Programa de Estudio y Control de Enfermedades Tropicales PECET, Calle 62 No. 52-59, Medellín, Colombia

## ABSTRACT

The Culicidae family has two of the most important disease vector genus: *Aedes* spp. and *Culex* spp. Both of these are involved in the transmission of arboviruses. Here, we provide novel data for the geographical distribution of 2,383 specimens in the Culicidae family. We also report the percentage of houses infested with these vectors, and Shannon and Simpson diversity indices in three municipalities located in Cauca, Colombia. This dataset is relevant for research on vector-borne diseases because *Aedes* spp. can transmit arboviruses such as dengue, Zika and chikungunya, and *Culex* spp. is a well-known vector of West Nile virus and Venezuelan equine encephalitis.

**Subjects**  Human and Biomedical Sciences, Global Health, Biomedical Science

**Submitted:**  28 February 2022

*  Corresponding author. E-mail: catalina.marcelo@udea.edu.co

Preprint submitted at https://doi.org/10.5281/zenodo.6525408

Included in the series: *Vectors of human disease* (https://doi.org/10.46471/GIGABYTE_SERIES_0002)

# DATA DESCRIPTION

## Introduction

Arboviruses like dengue, chikungunya, yellow fever and – more recently – Zika, are expanding, with more cases and fatalities, making them a concern for public health at the international level [1, 2]. In 2019, countries in the Americas recorded more than 2.7 million cases of dengue – the highest in history – including 22,127 severe cases and 1,206 deaths reported in October 2019 [3].

The main mosquitoes involved in the transmission of vector-borne disease are *Aedes aegypti* (Linnaeus), *Aedes albopictus* (Skuse) and *Culex quinquefasciatus* (Say). These belong to the Culicidae family (Diptera: Nematocera), which comprises about 3200 recognized species. Tropical rainforests, where fauna is more diverse but less well surveyed than temperate regions, probably house many more as-yet undiscovered species [4]. The *Ae. aegypti* mosquito is a predominantly domestic species, usually infesting natural or artificial containers in or around dwellings. The female feeds on human or domestic animal blood [5]. Owing to its close relationship with humans, it is essentially an urban mosquito. In Colombia, it has been recorded at altitudes ranging from 0–2302 m above sea level [6] and has also been reported in rural areas [7].

*Ae. aegypti* females restrict their hematophagous habits to daylight hours, with the peak of biting activity at dawn and shortly before dusk. Diurnal activity has also been recorded

in the male, where they copulate and feed on sugary substances. [8, 9]. Under optimal conditions of food availability and adequate oviposition sites, the average dispersion of a female *Aedes* spp. mosquito is estimated to be between 50 and 100 m, which limits its visits to two or three dwellings during its adult life [7, 10]. However, fed females have been recorded to disperse as far as 800 m in 6 days [11].

The mosquito *Ae. albopictus* (Skuse) is more commonly found in forested regions and in open spaces with abundant vegetation, typical of suburban or rural areas. They can be found in environments with extensive vegetation, such as bamboo stumps, tree cavities, plant axils (bromeliads) and water reservoirs in rock crevices as natural breeding sites [12]. When cohabiting with *Ae. aegypti* in urban areas, it can thrive in artificial reservoirs like flower vases, tires and cans [13]. However, although *Ae. albopictus* is anthropophilic, the hematophagous behavior of mosquito populations may depend on the availability of alternative food sources (domestic and wild animals) [14].

Finally, *Culex quinquefasciatus* (Say) is considered a highly anthropophilic species, which is associated with both urban and rural human habitats in Colombia [15]. Its biting activity occurs mostly nocturnally, but it remains indoors during the day [8, 16]. It is distributed across Colombia at elevations ranging from 0–3000 m above sea level (masl) [15]. Together with *Ae. aegypti*, the immature stages of *Cx. quinquefasciatus* also develop in structures serving as drains and catch-basins for stormwater located in streets and avenues [17].

Here we describe and quantify data associated with Culicidae adult mosquitoes (Diptera: Nematocera) collected with Prokopack aspirators in urban households in three municipalities with a high incidence of dengue in the department of Cauca (southwestern Colombia).

## Context

Data from the 2383 specimens (529 occurrences) reported here from the Culicidae family are novel unpublished data from the Cauca department, located in the Pacific region of Colombia. The data were collected in 2021 by a multidisciplinary team made up of environmental health technicians, geographic and environmental engineers, and professionals with extensive experience in medical entomology.

The mosquitoes were collected in three municipalities in the department of Cauca, an area of study chosen for its endemic–epidemic behavior for dengue disease. The event is characterized by focal endemics, variable transmission scenarios and temporal, seasonal and cyclical patterns in at-risk populations.

These municipalities are also part of an ongoing research project, which aims to determine the relationship between environmental, biological, and sociodemographic factors influencing the increase in the burden of dengue and its spatial variation, using geographic information systems in the construction of risk maps.

These data are valuable to the scientific community because they show the spatial location of specimen collection, allowing authorities to take action in those neighborhoods where mosquitoes have a higher percentage of house infestation. In addition, by being identified and geo-referenced, they allow various analyses concerning other entomological and environmental variables.

## METHODS

### Sampling

The collection of species belonging to the Culicidae family was done in the municipalities of Piamonte, Patía, and Miranda. These localities were chosen because they present a high risk of dengue according to incidence rate global cluster analysis (10,000 inhabitants), performed using the High-Low Clustering (Getis-Ord General) analysis in the ArcGIS® 10.8 software (SCR_011081). To detect hot spots, the incidence rates of dengue per 10,000 inhabitants from 2014–2018 were calculated from cases reported to the Sistema Nacional de Vigilancia en Salud Pública (SIVIGILA; National System of Public Health Surveillance), and the population statistics of the projection system reported by the National Administrative Department of Statistics in Colombia.

Afterwards, sampling was delimited to a neighborhood scale by Kernel density analysis (hot spot analysis) of dengue cases reported between 2015 and 2019 in the urban area of each municipality. The sample size was calculated through the estimated dengue prevalence (10.5%) in the municipality.

In total, 935 houses and 17 neighborhoods were visited during 2021. In Piamonte and Patía, three dengue clusters were identified, therefore three neighborhoods were sampled (*n* = 180 houses) for the first case, and seven neighborhoods (*n* = 335 houses) for the second. Finally, in Miranda, six neighborhoods (*n* = 420 houses) from four clusters with a higher-than-expected prevalence of the disease were sampled.

### Species collection

The entomological inspection was performed between 8:00 and 17:00 hours, in an average time of 10 minutes per house. In each house, a Prokopack aspirator (see Figure 1) was used to catch the adult mosquitoes present in the living room, dining room, bathrooms, kitchens, laundry yard and others, searching especially in shaded areas or near water containers. After the inspection was complete, collection cups were inspected to verify whether they had caught adult mosquitos. If so, then the collection cup was placed in a plastic bag along with a piece of chloroform-soaked cotton to immobilize the specimens, facilitating posterior differentiation and storage.

### Species classification and spatial characterization

After collecting the adult mosquitos, taxonomic identification and classification of the species *Ae. aegypti*, *Ae. albopictus* and *Cx. quinquefasciatus* was carried out by expert entomologists. Specialized taxonomic keys, such as those developed by Forattini [8] and Rueda [18] (Table 1) were used. Males were also distinguished from females.

Following the protocol for the fieldwork development [19], once females of the species *Ae. aegypti* were identified, they were stored in 0.2-ml vials for posterior processing using molecular biology techniques (Figure 2). Other species were stored dry for their subsequent entry into the collection by the Entomology group at Instituto Nacional de Salud (Bogota, Colombia).

Each vial was assigned a code corresponding to the house in which its specimens were collected. This code was associated with a sociodemographic survey applied to the same house using the application ArcGIS® Survey 123, which established the geographical position of the specimen collection. As well as metadata related to location, altitude values were obtained from the National Aeronautics and Space Administration (NASA) platform



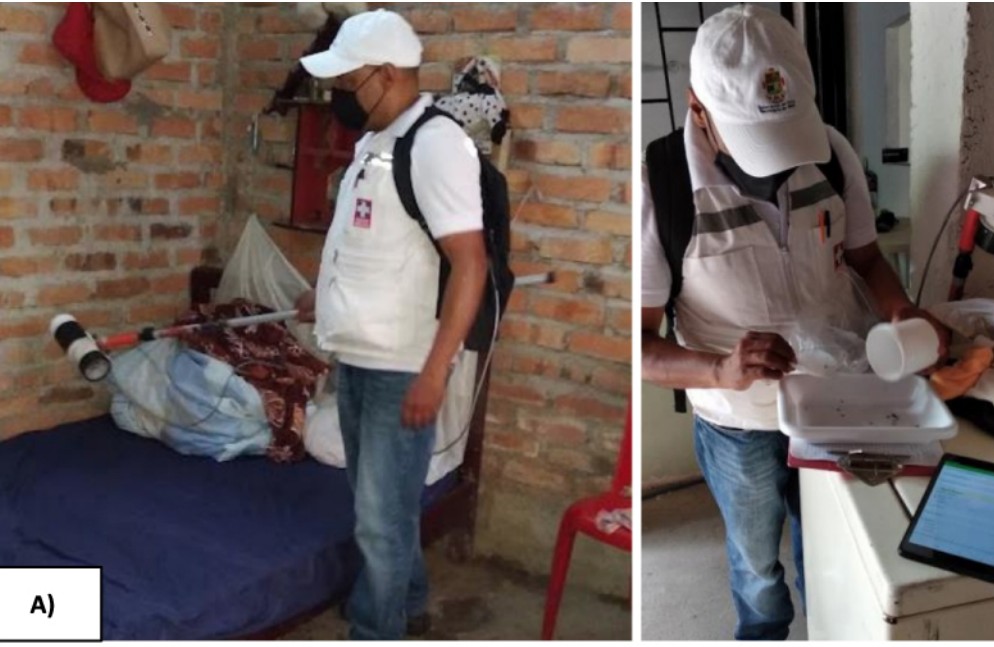

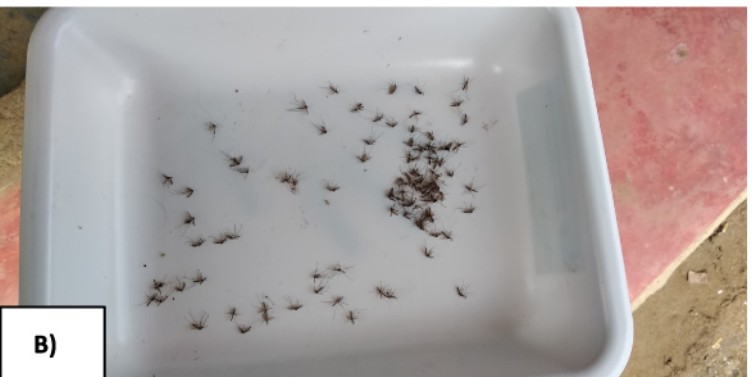

**Figure 1.** A staff member from the Cauca vector-borne diseases programme catches the mosquitoes using the Prokopack entomological aspirator. (A) Indoor capture and (B) mosquitoes collected in one of the sampled households.

**Table 1.** Descriptive entomological characteristics of the three species of mosquitoes recorded.

| Species | Taxonomic keys |
|---|---|
| *Aedes aegypti* (NCBI:txid7159) | Presence of lyre-shaped white longitudinal lines on the scutum. Clypeus with white scales (females). The scales in the mesanepimeron are separated and the tarsomeres of the posterior leg present basal white scales [7, 8, 18]. |
| *Aedes albopictus* (NCBI:txid7160) | Presence of only one white longitudinal line on the scutum. Clypeus has no scales (females) and scales from the mesanepimeron are 'V' shaped. Abdominal sclerites III–V are covered in dark scales [8]. |
| *Culex quinquefasciatus* (NCBI:txid7176) | Differs from *Aedes* for having a scutum with a brown integument (dark or light) densely covered in yellow scales [8]. |

ASF Data Search Vertex [20], which holds the global Digital Elevation Model (DEM) at a resolution of 12.5 m.

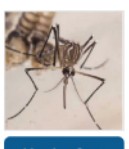

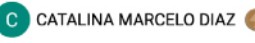

**Indoor active search for adult Aedes sp. and Culex sp. mosquitoes V.2** ▾

Erika Santamaria[1], Paola Muñoz[1], Catalina Marceló[1]
[1]Instituto Nacional de Salud (Bogotá, Colombia)

Version 2 ▾   dx.doi.org/10.17504/protocols.io.b5m7q49n

**C** CATALINA MARCELO DIAZ

**Protocols**   Materials   Metadata

ABSTRACT

This protocol details the indoor active search carried out in the research project: "Spatial stratification of dengue based on the identification of risk factors: a pilot trial in Cauca, Colombia"

Previous studies have reported that, indoors, adult *Aedes aegypti* mosquitoes rest on surfaces such as walls and furniture and generally at a height of less than 1.5 m (Dzul-Manzanilla et al., 2016 & Chadee, 2013). The use of electromechanical aspirators such as the Prokopack aspirator has also been reported (Vazquez-Prokopec *et al.*, 2009) which allows for a relatively easy collection of resting mosquitoes.

**Figure 2.** Protocols.io protocol for Indoor active search for adult *Aedes* sp. and *Culex* sp. mosquitoes [19]. https://www.protocols.io/widgets/doi?uri=dx.doi.org/10.17504/protocols.io.b5m7q49n

## DATA VALIDATION AND QUALITY CONTROL

A total of 2383 individuals corresponding to three species of the Culicidae family, *Ae. aegypti* ($n$ = 572), *Ae. albopictus* ($n$ = 36) and *Cx. quinquefasciatus* ($n$ = 1775) were collected in 529 of 935 houses located in Cauca, Colombia. Table 2 presents house infestation percentage per municipality, house index and additional descriptive measures of the sampled mosquitos at each location.

The number of adult *Ae. aegypti* and *Cx. quinquefasciatus* per house is shown as the percentage of total screened houses in Figure 3. It is worth highlighting that in three houses in Patía, more than 15 *Ae. aegypti* individuals were found, and the highest number of adults collected in a single house was 60. In Miranda, more than 10 individuals were found in four houses, and the highest number of adults in a single house was 20. Lastly, Piamonte presented the lowest number of adults per house, and the highest number in a single house was seven.

As a measure of community heterogeneity, Shannon and Simpson diversity indices were used [21]. The average diversity index for the three municipalities ranged from 0.27 to 0.66 for the Shannon index, and from 0.12 to 0.41 for the Simpson index. Both indices were highest in Patía and lowest in Piamonte. Graphically, location diversity can be better visualized using the Margalef index. For Miranda, the sample is very diverse, while for the other groups, a sample of diverse characteristics is observed. In general, these values are not high, being in the range of 0.28 to 0.35. (Figure 4).

The highest diversity index was recorded for *Ae. aegypti* (0.1488), followed by *Cx. quinquefasciatus* (0.0953), and least values were recorded in *Ae. albopictus* (0.0275). The dominance index of all mosquitoes sampled at three different municipalities was estimated (Table 3); the highest values were recorded in *Cx. quinquefasciatus* (0.5548) followed by *Ae. aegypti* (0.0576) and least in *Ae. albopictus* (0.0002). These results are similar to those of other previous studies [8].

Figure 5 shows the location and distribution map for Culicidae specimens collected in the urban area of the municipalities of Patía (Figure 5A), Miranda (Figure 5B) and Piamonte



**Table 2.** Descriptive entomological measures by sampling locality. The total of positive screened houses for each mosquito species is shown as a total and as a percentage for each municipality. Further information regarding sex-specific relative density and ratios females: males are also included.

| Entomological measure | Municipality | | | |
|---|---|---|---|---|
| | **Patia** | **Miranda** | **Piamonte** | **Total** |
| Number of houses screened | 335 | 420 | 180 | 935 |
| House index (HI) | 55.82 | 31.67 | 63.33 | 46.42 |
| Total number of Culicidae | 1156 | 301 | 926 | 2383 |
| Total number of *Ae. aegypti* (% of *Ae. aegypti*) | 305 (26.4) | 223 (74.1) | 44 (4.7) | 572 (24.0) |
| Number of positive houses with *Ae. aegypti* (% infested houses) | 102 (30.4) | 102 (24.3) | 28 (15.5) | 232 (24.8) |
| Number of *Ae. aegypti* females (% females) | 161 (52.8) | 118 (52.9) | 20 (45.4) | 299 (52.3) |
| Number of positive houses with *Ae. aegypti* females (% of positive houses) | 73 (71.6) | 75 (73.5) | 15 (53.6) | 163 (70.2) |
| Number of *Ae. aegypti* males (% of males) | 144 (47.2) | 105 (47.1) | 24 (54.5) | 273 (47.7) |
| Number of positive houses with *Ae. aegypti* males (% of positive houses) | 63 (61.8) | 56 (54.9) | 17 (60.7) | 136 (58.6) |
| *Ae. aegypti* sex ratio F:M | 1.1:1 | 1.1:1 | 0.8:1 | 1.1:1 |
| Total number of *Cx. quinquefasciatus* (% of *Cx. quinquefasciatus*) | 831 (71.9) | 76 (25.2) | 868 (93.7) | 1175 (74.5) |
| Number of positive houses with *Cx. quinquefasciatus* (% infested houses) | 121 (36.1) | 49 (11.7) | 108 (60.0) | 278 (29.7) |
| Number of *Cx. quinquefasciatus* females (% of females) | 190 (22.9) | 53 (69.7) | 374 (43.1) | 617 (34.8) |
| Number of positive houses with *Cx. quinquefasciatus* females (% of positive houses) | 66 (54.5) | 39 (79.6) | 85 (78.7) | 190 (68.3) |
| Number of *Cx. quinquefasciatus* males (% of males) | 641 (77.1) | 23 (30.3) | 494 (56.9) | 1158 (65.2) |
| Number of positive houses with *Cx. quinquefasciatus* males (% of positive houses) | 105 (86.8) | 13 (26.5) | 91 (84.2) | 209 (75.2) |
| *Cx. quinquefasciatus* sex ratio F:M | 0.3:1 | 2.3:1 | 0.7:1 | 0.5:1 |
| Total number of *Ae. albopictus* (% of *Ae. albopictus*) | 20 (1.7) | 2 (0.7) | 14 (1.5) | 36 (1.5) |
| Number of positive houses with *Ae. albopictus* (% infested houses) | 20 (6.0) | 2 (0.5) | 5 (2.8) | 27 (2.9) |
| Number of *Ae. albopictus* females (% females) | 9 (45.0) | 2 (100.0) | 8 (57.1) | 19 (52.8) |
| Number of positive houses with *Ae. albopictus* females (% of positive houses) | 8 (40.0) | 2 (100.0) | 5 (100.0) | 15 (55.6) |
| Number of *Ae. albopictus* males (% of males) | 11 (55.0) | 0 (0.0) | 6 (42.9) | 17 (47.2) |
| Number of positive houses with *Ae. albopictus* males (% of positive houses) | 5 (25.0) | 0 (0.0) | 2 (40.0) | 7 (25.9) |
| *Ae. albopictus* sex ratio F:M | 0.8:1 | 2.0:0 | 1.3:1 | 1.1:1 |

**Table 3.** Species richness and diversity indices of mosquitoes found in the department of Cauca, Colombia.

| | $f_i$ | $f_i \log f_i$ | $f_i \log^2 f_i$ | $P_i$ | $N_i (n_i-1)/n (N-1)$ | $P_i \log P_i$ | $P_i \ln P_i$ | $P_i (\ln P_i)2$ | Shannon Weiner Index $H = (N \log N - \Sigma\ f_i \log f_i/N)$ (or) − $(P_i \log P_i)$ | Simpson's Dominance Index $C = \Sigma$ |
|---|---|---|---|---|---|---|---|---|---|---|
| *Ae. aegypti* | 572 | 1577.2305 | 4349.0492 | 0.2400 | 0.0576 | −0.1488 | −0.3425 | 0.4888 | 0.1488 | 0.0576 |
| *Ae. albopictus* | 36 | 56.0269 | 87.1948 | 0.0151 | 0.0002 | −0.0275 | −0.0633 | 0.2655 | 0.0275 | 0.0002 |
| *Cx.quinquefasciatus* | 1775 | 5767.3271 | 18739.1897 | 0.7449 | 0.5548 | −0.0953 | −0.2194 | 0.0646 | 0.0953 | 0.5548 |
| Σ | 2383 | 7400.5845 | 23175.4337 | 1 | 0.6127 | −0.2716 | −0.6253 | 0.81895 | 0.2716 | 0.6127 |

(Figure 5C), differentiated by genus and number of specimens per household. The taxonomic category *Aedes* includes the dengue-transmitting species *Ae. aegypti* and *Ae. albopictus*, while the *Culex* category includes the species *Cx. quinquefasciatus*, which is also involved in the transmission of arbovirosis.

In all the mosquito catches, previously designed e-formats designed in Survey123 (ESRI) were used to record field collection information and household geographic coordinates. Expert entomologists advised taxonomical identification. Dataset records were confirmed and verified individually.

Only *Aedes* spp. and *Cx. quinquefasciatus* were collected with the sampling method used (Prokopack aspirators). For future studies, complementary sampling methods or monitoring techniques should be used for adult mosquitoes (for example, ovitraps, BG-SENTINEL mosquito traps).

Before a specimen is deposited in the collection, collection curators from the Entomology Group review the information associated with the specimen; that is, the locality,



**Figure 3.** Percentage of houses with their corresponding number of *Ae. aegypti* and *Cx. quinquefasciatus* adults. The total percentage shown corresponds to the total screened houses for each municipality.

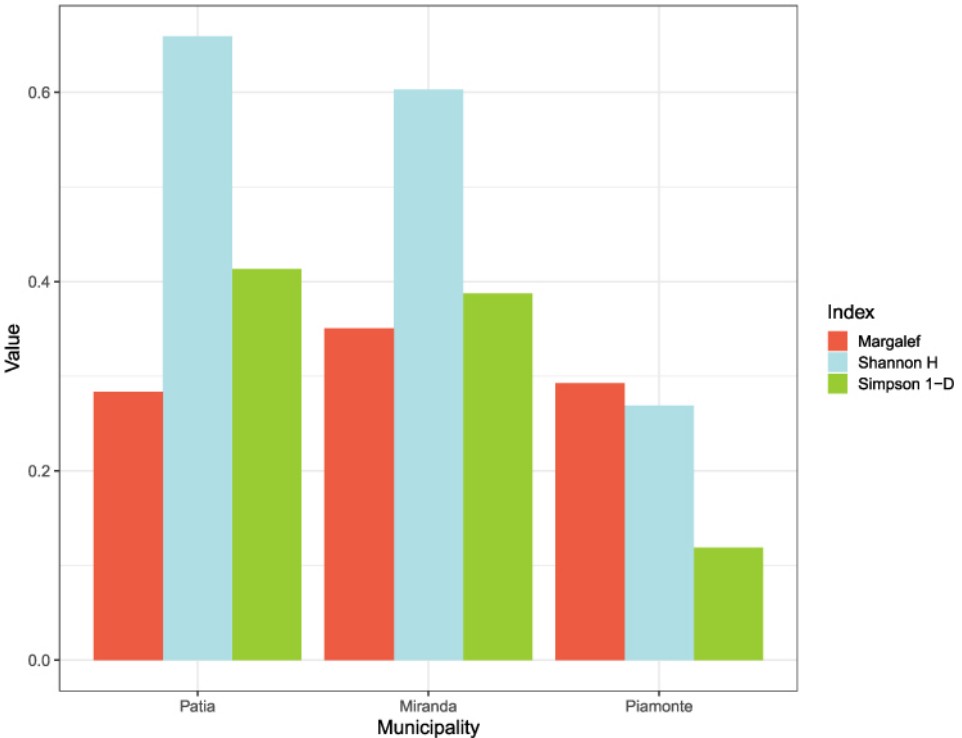

**Figure 4.** Species diversity measures for each municipality. Shannon Index is represented by blue bars, Simpson Index is represented by green bars and Margalef Index is represented by red bars.

geographical coordinates, sex, stage of development, and taxonomy. The minimum information required to include a specimen in the collection is related to the standard Darwin Core and is the same as the minimum information required for publication in the Global Biodiversity Information Facility (GBIF).

## RE-USE POTENTIAL

The database and vector distribution map provide a novel resource for understanding the abundance and behavior of mosquito-borne diseases for the entomology and vector-borne diseases community. To improve the accessibility and usability of these data, they have been included in the GBIF.

These data will be useful for museums setting up similar displays, and the data may be used for other research purposes such as predictive models and species geographic distributions. The *Ae. aegypti* and *Ae. albopictus* database may also be helpful for similar projects elsewhere. We suggest to others to make their data similarly available.

## DATA AVAILABILITY

The datasets supporting this article are available in the GBIF repository [22].

## EDITOR'S NOTE

This paper is part of a series of Data Release articles working with GBIF and supported by TDR, the Special Programme for Research and Training in Tropical Diseases, hosted at the World Health Organization [23].

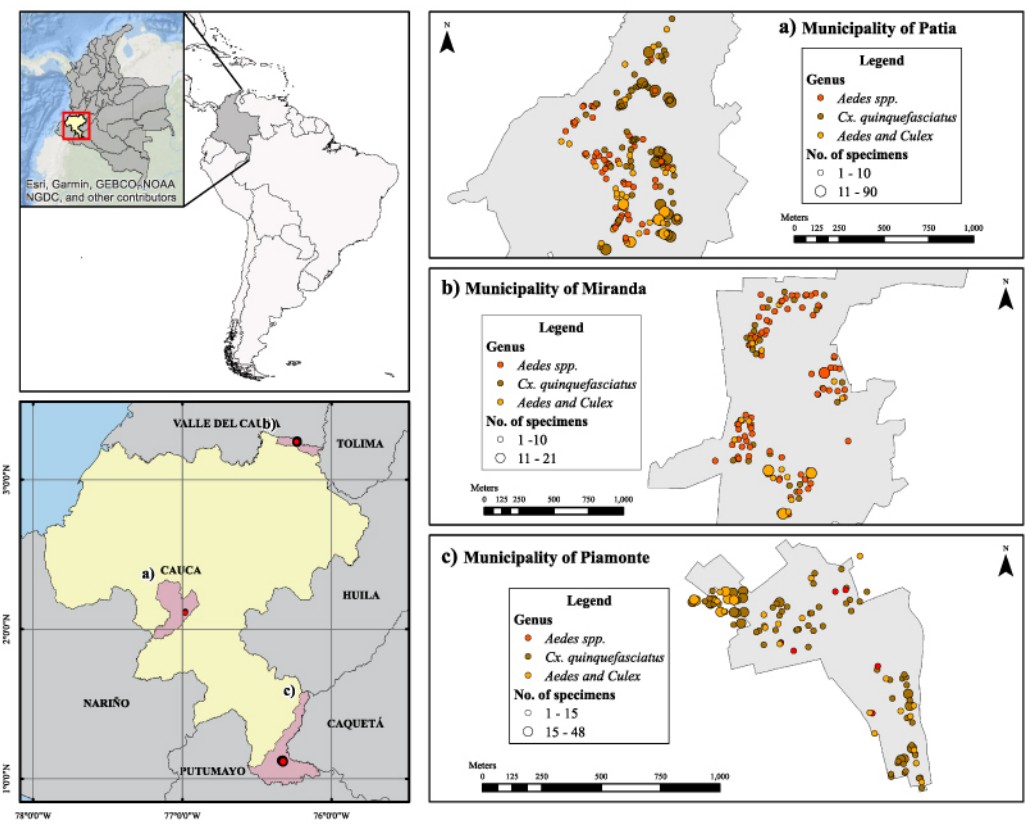

**Figure 5.** Distribution and density for Culicidae specimens collected in the urban area of the municipalities of the Cauca department.

## DECLARATIONS
## LIST OF ABBREVIATIONS

DEM: digital elevation model; ESRI: Environmental Systems Research Institute; GBIF: Global Biodiversity Information Facility; NASA: National Aeronautics and Space Administration; SIVIGILA: Sistema Nacional de Vigilancia en Salud Pública (National System of Public Health Surveillance).

## ETHICS APPROVAL AND CONSENT TO PARTICIPATE

The Ethics and Research Methodology Committee (CEMIN) of the Instituto Nacional de Salud, created by resolution 395 of 4 April 2017, considers that the research project "Spatial stratification of dengue based on the identification of risk factors: a pilot trial in the department of Cauca" (CEMIN 13-2019), meets the technical and ethical requirements for which it is granted approval according to Act 11 of 23 May 2019.

## CONSENT FOR PUBLICATION

Not applicable.

## COMPETING INTERESTS

The authors declare that they have no competing interests.

## FUNDING

This work was supported by the National Institute of Health, Bogotá, Colombia (research project CEMIN 13-2019), the Secretaria de Salud Departamental del Cauca, the Universidad de Ciencias Aplicadas y Ambientales UDCA and Ministerio de Ciencia, Tecnología e Innovación de Colombia (research project 210484467217).

## AUTHORS' CONTRIBUTIONS

CM and ES conceptualised the study, with CAM made the project administration. CM, ES, MCL, and SM drafted the manuscript. CAM, PF, ES and MCL generated the field data. PF, HC, CAM and AAD reviewed and edited the draft. All authors made comments on the manuscript.

## ACKNOWLEDGEMENTS

We thank officials of the Department of Health of Cauca, especially the leader of the Inspection, Surveillance and Health Control Process, Hernando Gil Gómez, for his support in the realization of this research project; to the engineer Anderson Hair Piamba in the ETV Program for his support; and to the vector-borne diseases technicians in the municipality of Piamonte, Patía and Miranda for *Ae. aegypti* catches. We thank Marco Fidel Suárez for his valuable assistance in performing the fieldwork and knowledge.

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
