## [Reviewer Report]

Upload additional filesDRR-202202-04/form/GigaByte_Review_DRR_202202_04_Arbovirus_CJA_08Mar2022.pdfReviewer name and names of any other individual's who aided in reviewer Chris ArmitDo you understand and agree to our policy of having open and named reviews, and having your review included with the published papers. (If no, please inform the editor that you cannot review this manuscript.)YesIs the language of sufficient quality?NoPlease add additional comments on language quality to clarify if needed
The GBIF Dataset page and the Metadata are in Spanish, and so I cannot determine the quality of the language in these resources.Are all data available and do they match the descriptions in the paper? YesAdditional CommentsAre the data and metadata consistent with relevant minimum information or reporting standards? See GigaDB checklists for examples <a href="http://gigadb.org/site/guide" target="_blank">http://gigadb.org/site/guide</a>YesAdditional CommentsIs the data acquisition clear, complete and methodologically sound?YesAdditional CommentsIs there sufficient detail in the methods and data-processing steps to allow reproduction?YesAdditional CommentsIs there sufficient data validation and statistical analyses of data quality? YesAdditional CommentsIs the validation suitable for this type of data?YesAdditional CommentsIs there sufficient information for others to reuse this dataset or integrate it with other data?YesAdditional CommentsAny Additional Overall Comments to the AuthorThis Data Release manuscript describes the occurrence of mosquitoes in Colombia. The dataset is in Spanish, and the Metadata consists of 529 Geographic Locations and include the different Latitude / Longitude coordinates where Sampling was accomplished. These Geographic Locations relate to the Colombian municipalities of: 1) Patia; 2) Miranda, and; 3) Piamonte, which are all based in Cauca which is a Department in Southwestern Colombia.

The GBIF Metadata with Latitude / Longitude coordinates was included in the Darwin Core DwC package “occurrence.txt” file. As a reviewer, I performed a spot check of 10 Samples, and I can confirm that the Latitude / Longitude coordinates for these 10 Samples refer to the aforementioned locations. There are a total of 3 mosquito species referred to in the Metadata – Aedes albopictus, Aedes aegypti, Culex quinquefasciatus – which is consistent with the manuscript. In addition, the manuscript states the following:

• “A total of 2,383 individuals corresponding to three species of the Culicidae family: Ae. Aegypti (n=572), Ae. albopictus (n=36) and Cx. quinquefasciatus (n=1,775)”

The values in the GBIF Dataset are in agreement with this statement. Furthermore, the manuscript refers to 529 occurrences, and so GBIF Metadata is also consistent with this.

The major issues with this Dataset are as follows:

1. The GBIF Dataset is in Spanish.

2. The License is: CC-BY-NC 4.0

3. The DOI redirects to the IPT webpage rather than the GBIF Dataset page 

• https://doi.org/10.15472/dxbowv

4. There is a link in the IPT Dataset that I am not authorised to access:

• https://ipt.biodiversidad.co/cr-sib/resource.do?r=01300_ins-dipteroscauca_20220128

RecommendationMinor Revision

---

## [Reviewer Report]

Reviewer name and names of any other individual's who aided in reviewer Nils TjadenDo you understand and agree to our policy of having open and named reviews, and having your review included with the published papers. (If no, please inform the editor that you cannot review this manuscript.)YesIs the language of sufficient quality?YesPlease add additional comments on language quality to clarify if needed
Are all data available and do they match the descriptions in the paper? YesAdditional Commentsavailable on GBIF, which is exactly the right place for itAre the data and metadata consistent with relevant minimum information or reporting standards? See GigaDB checklists for examples <a href="http://gigadb.org/site/guide" target="_blank">http://gigadb.org/site/guide</a>YesAdditional Commentsfollows GBIF metadata standards, which appear to be in line with what GigaDB expectsIs the data acquisition clear, complete and methodologically sound?NoAdditional CommentsSelection of study sites could be clearer: Where did the incidence data come from? Where did the case numbers for the neighborhoods come from? Were all houses in a selected neighborhood visited or just a selection? Were all visited houses accessible to sampling?Is there sufficient detail in the methods and data-processing steps to allow reproduction?YesAdditional CommentsThis is pretty much unprocessed, raw data.Is there sufficient data validation and statistical analyses of data quality? YesAdditional CommentsReally not much to validate here. Locations seem to be located where they are supposed to be, and apparently all records were double-checked individually. It's already been through the GBIF system too.

In terms of (explorative) statistics, I was wondering why Figure 2 was limited to one of 3 species.Is the validation suitable for this type of data?YesAdditional CommentsIs there sufficient information for others to reuse this dataset or integrate it with other data?YesAdditional CommentsAny Additional Overall Comments to the AuthorDear Authors,
Thanks not only for sharing this data set but also for making it CC-0!

Some minor points:

Reference #1 does not appear to suitable to support the claim made in the first sentence of the introduction, at least not on its own. The sentence is about global patterns of several diseases, the cited paper by Ye & Moreno is about dengue in Colombia only. Maybe it was supposed to go with the following sentence? In any case, there must be a suitable review or WHO report you could cite instead.

First sentence of second paragraph of introduction: 1) "y" -> "and" 2) there appears to be a "which" missing before "consists of about 3,200 recognized species".

Figure 1 is not referenced in the main manuscript text.

Figure 3, upper left corner: some text has been cut off (suppose "sri" should be "Esri"?)RecommendationMinor Revision

---

## [Reviewer Report]

Upload additional filesDRR-202202-04/form/Main text (1).docxReviewer name and names of any other individual's who aided in reviewer Dr. P. Senthamarai SelvanDo you understand and agree to our policy of having open and named reviews, and having your review included with the published papers. (If no, please inform the editor that you cannot review this manuscript.)YesIs the language of sufficient quality?YesPlease add additional comments on language quality to clarify if needed
In the introduction part try to avoid too many split in paragraph. And also minimize the paragraph with coherence. Are all data available and do they match the descriptions in the paper? YesAdditional CommentsAre the data and metadata consistent with relevant minimum information or reporting standards? See GigaDB checklists for examples <a href="http://gigadb.org/site/guide" target="_blank">http://gigadb.org/site/guide</a>YesAdditional CommentsIs the data acquisition clear, complete and methodologically sound?NoAdditional CommentsAuthors should use statistical tools for representing data.Is there sufficient detail in the methods and data-processing steps to allow reproduction?NoAdditional CommentsAuthors have taken extra care to process the identification of Culicidae, but still, there is a chance for escape while identification. How they will manage. Is there sufficient data validation and statistical analyses of data quality? NoAdditional CommentsThe authors were not used sufficient statistical tools to interpret the data. They proved numerically but not statistically. Ref: 10.1016/j.actatropica.2016.03.029Is the validation suitable for this type of data?NoAdditional CommentsAuthors should use indices for this study. Try to use the House index, Container index and Breteau index.
Refer: Senthamarai Selvan and Jebanesan, Indian Journal of Natural Products and Resources Vol. 7(3), September 2016, pp. 234-239Is there sufficient information for others to reuse this dataset or integrate it with other data?YesAdditional CommentsI appreciate the investigators for getting such meaningful data.Any Additional Overall Comments to the AuthorAuthors should explain, other species while collecting the mentioned vectors viz., Aedes aegypti, Aedes albopictus and Culex quinquefasciatus. As per the statement, the authors collected only those three species, but there is a high chance for the collection of other species also. Why the authors are not considering/mentioned on other species.RecommendationMajor Revision